# Differential Influence of *Pueraria lobata* Root Extract and Its Main Isoflavones on Ghrelin Levels in Alcohol-Treated Rats

**DOI:** 10.3390/ph15010025

**Published:** 2021-12-24

**Authors:** Michał Szulc, Radosław Kujawski, Justyna Baraniak, Małgorzata Kania-Dobrowolska, Ewa Kamińska, Agnieszka Gryszczyńska, Kamila Czora-Poczwardowska, Hanna Winiarska, Przemysław Ł. Mikołajczak

**Affiliations:** 1Department of Pharmacology, Poznań University of Medical Sciences, Rokietnicka 5a, 60-806 Poznan, Poland; radkuj@ump.edu.pl (R.K.); awekam@ump.edu.pl (E.K.); kczora@ump.edu.pl (K.C.-P.); hwiniar@ump.edu.pl (H.W.); przemmik@ump.edu.pl (P.Ł.M.); 2Department of Pharmacology and Phytochemistry, Institute of Natural Fibres and Medicinal Plants, Kolejowa 2, 62-064 Plewiska, Poland; justyna.baraniak@iwnirz.pl (J.B.); malgorzata.kania@iwnirz.pl (M.K.-D.); agnieszka.gryszczynska@iwnirz.pl (A.G.)

**Keywords:** acamprosate, daidzin, naltrexone, puerarin, *Pueraria lobata*, alcohol intake, alcohol preferring rats, alcohol tolerance, ghrelin blood level

## Abstract

The study was carried out on alcohol-preferring male Wistar rats. The following drugs were repeatedly (28×) administered: acamprosate (500 mg/kg, p.o.), naltrexone (0.1 mg/kg, i.p), and *Pueraria lobata* (kudzu) root extract (KU) (500 mg/kg, p.o.) and its isoflavones: daidzin (40 mg/kg, p.o.) and puerarin (150 mg/kg, p.o.). Their effects on a voluntary alcohol intake were assessed. KU and alcohol were also given for 9 days in an experiment on alcohol tolerance development. Finally, total and active ghrelin levels in peripheral blood serum were measured by ELISA method. Acamprosate, naltrexone, daidzin, and puerarin, reducing the alcohol intake, caused an increase in both forms of ghrelin levels. On the contrary, though KU inhibited the alcohol intake and alcohol tolerance development, it reduced ghrelin levels in alcohol-preferring rats. The changes of ghrelin concentration could play a role as an indicator of the currently used drugs. The other effect on the KU-induced shift in ghrelin levels in the presence of alcohol requires further detailed study.

## 1. Introduction

Excessive alcohol consumption is a community-wide problem and can lead to many diseases [1]. Alcohol abuse has been on an upward trend over the past two years, most likely fueled by the COVID-19 epidemic [2]. The effects of excessive alcohol consumption include increased healthcare costs, alcohol-related crimes (e.g., assault and robbery), and motor vehicle accidents [3]. Long-term treatment of alcoholism is primarily based on detoxification; the patient undergoing psychotherapy is additionally supported by drugs that facilitate the rejection of alcohol [4,5,6]. Due to the lack of sufficiently effective drugs in this field, searching for new agents, including plant-derived substances that reduce the amount of alcohol consumed, is extremely promising [7,8].

*Pueraria lobata* (Willd) Ohwi (kudzu, kudzu vine) is a plant [9] native to eastern Asia, possessing a long history of traditional usage as medicine in Asian countries, with roots as the main raw material for active substances, among others, flavonoids (isoflavones), isoflavone glucosides, organic acids, saponins, starch, and D-mannitol [9]. Three compounds from the group of isoflavones deserve particular attention, namely: daidzin (DAI), daidzein, and puerarin (PUE). *P. lobata* preparations and their main components have potential in an alcohol intake reduction [10], and thus in the long-term prevention of alcohol addiction, has been indicated in a number of studies over the past two decades [11,12,13]. For instance, DAI and daidzein effectively inhibited ethanol (EtOH) consumption in laboratory animals [10,14]. PUE has been also proved to dose-dependently reduce alcohol consumption [15] and to suppress alcohol withdrawal symptoms. The ability of PUE to decrease the amount of beer consumption and the increase in the time course of drinking was also proved in a clinical trial [16]. Several other clinical examinations provided further evidence of the ability to inhibit the amount or rate of alcohol consumption [17,18,19,20]. In addition to suppressing alcohol intake, plant remedies from kudzu also exhibit other interesting pharmacological activities, e.g., preventing obesity and improving glucose metabolism [21,22,23]. Other preclinical studies indicated their ability to ameliorate glucose and lipid metabolic disorders [24,25]. Anti-inflammatory and antioxidant activities of kudzu root extract (especially in the case of PUE) have also been demonstrated [26], and similar properties were observed in the leaves [27]. It is worth mentioning that PUE, the main constituent isolated from *P. lobata* roots, also revealed a beneficial effect on aging related diseases [9] and has received investigational drug status for the treatment of alcohol abuse as well as being listed in the DRUGBANK database [28].

A crucial part of the reward system in the brain is the cholinergic–dopaminergic pathway [29,30], which is strongly associated with the reinforcing properties of rewards [29,31]. The influence of drug dependence on the reward system has been widely described for ghrelin and its receptor, GHS-R1A [30,32,33,34,35]. In general, ghrelin presence in the brain has been well documented [30,36] and the active form of ghrelin passes the blood-brain barrier [37], even when released or administered peripherally [38], and in this way, it may have central effects [30,39]. There are some studies on the role of GHS-R1A [40,41], which is down-regulated in the ventral tegmental area (VTA) in alcohol high-preferring rats, compared to low-alcohol consuming rats [40]. Several studies with the usage of GHS-R1A antagonists (i.e., JMV2959, BIM28163 or D-Lys3-GHRP-6) strongly supported the assumption stating that ghrelin signaling regulates alcohol intake in high-alcohol consuming Wistar rats, as well as in alcohol-preferring (AA) rats [40,42,43]. Moreover, the alcohol reward disruption and the alcohol-induced locomotor stimulation and dopamine release in nucleus accumbens (NAc) was observed in ghrelin knockout mice [44,45]. Interestingly, the nature of the interaction processes between ghrelin and GHS-R1a receptors, documented particularly in appetite signaling and closely related to reward behavior [34], has an ability to dimerize with several additional G protein-coupled receptors (GPCRs), including the GHS-R1b receptor [46]. Their molecular mechanism of action during EtOH addiction development remains unclear [36].

Interesting observations were made in the course of the analysis of changes in the peripheral concentration of ghrelin, both in rodents [47,48,49] and in clinical trials [50,51,52,53] under conditions of significant ethanol consumption. Results from clinical observations support data showing that active drinking in alcohol-dependent individuals may suppress both ghrelin levels and its production [50,51,52], observed in our previous study on high-alcohol preferring rats as well [47]. Active form of ghrelin (acyl-ghrelin) elevation by EtOH during early adolescence was observed [48]. Experiments in model animals (two-bottle choice protocol) indicate that the key role of the ghrelin signaling pathway in lowering of alcohol voluntary intake appears to be equal for both sexes [49].

With the above in mind, the aim of this study was to determine the role of *P. lobata* root extract (KU) and its main isoflavones (PUE, DAI) on ghrelin blood level in the experimental model of alcohol dependence. It is the quest for a correlation between the ability of KU to reduce alcohol drinking behavior using free choice procedure and the concentration of both forms of ghrelin in rats. Furthermore, the effect of KU on alcohol tolerance development coupled with ghrelin blood level was an additional research target. The influences of KU, PUE, and DAI on ghrelin levels were compared to the effects of standard drugs used to inhibit the alcohol drinking reflex, i.e., acamprosate (AC) and naltrexone (NAL). The chemical structures of these compounds are presented in Figure 1.

## 2. Results

### 2.1. Content of Isoflavones in KU

The HPLC-DAD chromatogram of the isoflavones contained in KU is presented in Figure 2.

Isoflavone contents determined by the KU chromatogram are gathered in Table 1.

It was found that PUE constituted 13.3% of the compounds tested, while its derivatives 3′-hydroxypuerarin, 3′-methoxypuerarin and 6″-O-D-xylosylpuerarin measured approximately four times smaller (2.31%, 2.89%, and 2.55%, respectively). The remaining isoflavones, i.e., DAI and daidzein, were rated 2.88% and 0.74%, respectively.

**Table 1 pharmaceuticals-15-00025-t001:** Content of isoflavones in KU.

No.	Compound	Concentration [mg/g]
I	3′−hydroxypuerarin	23.1 ± 2.0
II	puerarin	133.3 ± 11
III	3′−methoxypuerarin	28.9 ± 3.0
IV	6″−O−D−xylosylpuerarin	25.5 ± 3.0
V	daidzin	28.8 ± 5.0
VI	daidzein	7.40 ± 3.0

Mean ± SD (*n* = 5).

### 2.2. KU Repeated Administration Effects on Alcohol Drinking Behavior and Ghrelin Levels

In this experiment, the effect of repeated intragastric (p.o.) KU administration (28×) at a dose of 500 mg/kg on the amount of EtOH intake by Wistar rats (*n* = 36) was observed. The rats were previously divided into alcohol-preferring (PR; *n* = 18) and non-preferring (NP; *n* = 18) groups. Subsequently, they were divided randomly into four groups containing nine animals each (NP_MC, NP_KU, PR_MC, PR_KU).

Based on the obtained data on drinking behavior, there was a significant variability between the groups (ANOVA: F(3,28) = 86.0; *p* = 0.0000). Further statistical analysis showed that control preferring animals (PR_MC) drank much more alcohol than control non-preferring rats (NP_MC) (*p* < 0.01) (Figure 3A). The administration of KU significantly reduced the amount of EtOH consumed by alcohol preferring rats (PR_KU) in relation to the PR_MC group (*p* < 0.001). A similar significant effect was observed in non-preferring animals (NP_KU) in relation to NP_MC (*p* < 0.01) (Figure 3A).

The conducted experiment did not significantly affect the condition of the animals (Figure 3B) as there was no significant variation in their body mass (ANOVA: F(3,28) = 0.92; *p* = 0.4414) nor was there any difference in the amount of total fluid intake (ANOVA: F(3,28) = 0.65; *p* = 0.5857).

In this experiment, the effect of KU on ghrelin level was also examined in PR and NP groups. There was significant intergroup variability both for active ghrelin concentrations (ANOVA: F(3,28) = 5.11; *p* = 0.0060) and for total ghrelin levels (ANOVA: F(3,28) = 4.96; *p* = 0.0069). Further statistical analysis showed that in the animals preferring to drink EtOH, the level of active ghrelin was significantly reduced (PR_MC vs. NP_MC), *p* < 0.05) (Figure 3C). In preferring animals, after KU administration, even lower concentrations of active ghrelin were observed (PR_KU vs. PR_MC, *p* < 0.05) (Figure 3C). Similarly, the level of total ghrelin was lower in PR rats when compared with NP animals (PR_MC vs. NP_MC, *p* < 0.05) (Figure 3D). However, KU administration to preferring animals decreased the levels of total ghrelin in blood, but the difference between the obtained values for the study group and the control animals did not reach statistical significance (PR_KU vs. PR_MC, *p* > 0.05). Such an effect was not found in NP rats (NP_KU vs. NP_MC, *p* > 0.05) (Figure 3D).

### 2.3. DAI Repeated Administration Effects on Alcohol Drinking Behavior and Ghrelin Levels

In this experiment, the effect of repeated (28×) intragastric (p.o.) administration of DAI at a dose of 40 mg/kg on the amount of EtOH consumed in Wistar rats (*n* = 36) was observed. The rats were previously divided into alcohol-preferring (PR; *n* = 18) and non-preferring (NP; *n* = 18) groups. Later, they were divided randomly into four groups containing nine animals each (NP_MC, NP_DAI, PR_MC, PR_DAI). Based on the obtained data on drinking behavior, there was a significant variability between the groups (ANOVA: F(3,34) = 7.13; *p* = 0.0008). Further analysis showed that PR_MC animals drank significantly more alcohol compared to NP_MC rats (*p* < 0.001) (Figure 4A). The administration of DAI to PR rats (PR_DAI) resulted in a significant reduction in the amount of EtOH consumed in relation to the proper control group (PR_MC) (*p* < 0.001) (Figure 4A). However, there was no effect of DAI on alcohol consumption in NP animals (NP_DAI vs. NP_MC, *p* > 0.05).

The experiment did not significantly affect the condition of the animals (Figure 4B), because there was no significant variation in the weight of the animals (ANOVA: F(3,34) = 1.64; *p* = 0.1969), nor was there any difference in the amount of total fluid intake (ANOVA: F(3,34) = 1.15; *p* = 0.3450).

The effect of DAI on ghrelin levels in NP and PR rats was also investigated. There was a significant variability in the levels of both active (ANOVA: F(3,34) = 6.49; *p* = 0.0015) and total ghrelin (ANOVA: F(3,34) = 9.41; *p* = 0.0001). Further statistical analysis showed that PR rats (PR_MC) had much lower concentrations (*p* < 0.001) of both active (Figure 4C) and total (Figure 4D) ghrelin compared to NP (NP_MC) animals. The administration of DAI to PR rats increased the concentrations of both forms of ghrelin (*p* < 0.001), leading to the values observed in NP_MC rats (Figure 4C,D). However, the isoflavonoid did not change the concentration of both total and active ghrelin in NP animals.

### 2.4. PUE Repeated Administration Effects on Alcohol Drinking Behavior and Ghrelin Levels

In this experiment, the effect of repeated (28×) intragastric (p.o.) administration of PUE at a dose of 150 mg/kg on the amount of EtOH drank in Wistar rats (*n* = 44) was observed. The rats were previously divided into preferring alcohol drinking (PR; *n* = 22) and non-preferring (NP; *n* = 22) groups. Later, they were randomly divided into four groups containing 11 animals each (NP_MC, NP_PUE, PR_MC, PR_PUE).

On the basis of the obtained results, a statistically significant variability between the groups in the amount of drank EtOH was observed (ANOVA: F(3,40) = 25.1; *p* = 0.0000). Further analysis showed that PR_MC animals consumed much more alcohol compared to NP_MC animals (*p* < 0.001) (Figure 5A). The administration of PUE to PR (PR_PUE) rats resulted in a significant reduction in the amount of EtOH consumed in relation to the PR_MC group (*p* < 0.001) (Figure 5A). However, no effect of PUE on alcohol consumption in NP animals was found (NP_PUE vs. NP_MC, *p* > 0.05).

**Figure 4 pharmaceuticals-15-00025-f004:**
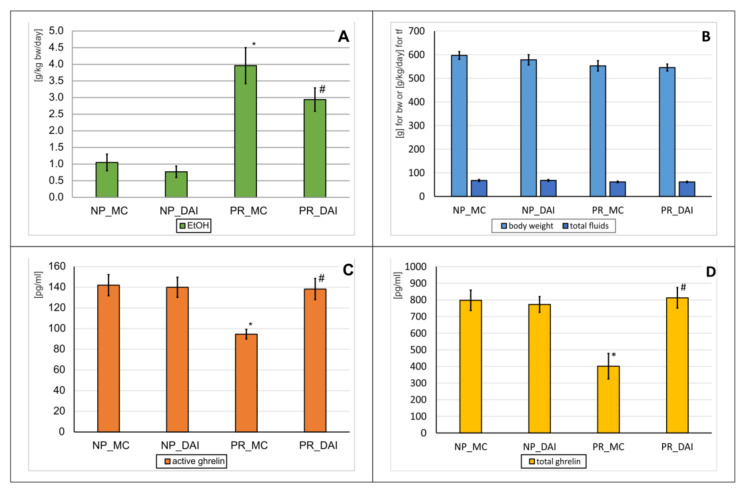
The effect of repeated (28×) daidzin (DAI) (40 mg/kg, p.o.) administration on alcohol drinking behavior, total fluid intake, body weight, and ghrelin levels in preferring (PR) and non-preferring (NP) rats. Legend: *n* = 8–9; mean ± SEM; PR_MC—control PR rats; NP_MC—control NP rats; MC—vehicle (0.5% methylcellulose); (**A**) amount of drunk ethanol [g/kg/day]; *—vs. NP_MC group, *p* < 0.001; #—vs. PR_MC group, *p* < 0.001; (**B**) body weight [g] and the total amount of fluid intake [g/kg/day]; (**C**) serum active ghrelin levels [pg/mL]; *—vs. NP_MC group, *p* < 0.001; #—vs PR_MC, *p* < 0.001; (**D**) serum total ghrelin levels [pg/mL]; *—vs. NP_MC group, *p* < 0.001; #—vs PR_MC, *p* < 0.001.

**Figure 5 pharmaceuticals-15-00025-f005:**
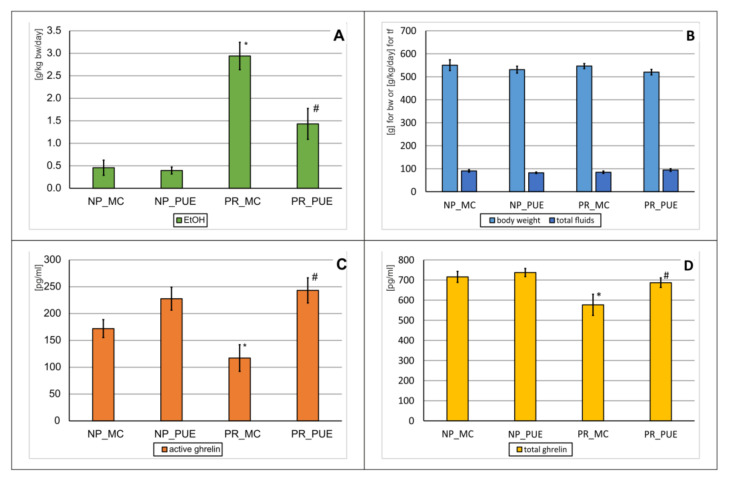
The effect of repeated (28×) puerarin (PUE) (150 mg/kg, p.o.) administration on alcohol drinking behavior, total fluid intake, body weight, and ghrelin levels in preferring (PR) and non-preferring (NP) rats. Legend: *n* = 9–11; mean ± SEM; PR_MC—control PR rats; NP_MC—control NP rats; MC—vehicle (0.5% methylcellulose); (**A**) amount of drunk ethanol [g/kg/day]; *—vs. NP_MC group, *p* < 0.001; #—vs. PR_MC group, *p* < 0.001; (**B**) body weight [g] and the total amount of fluid intake [g/kg/day]; (**C**) serum active ghrelin levels [pg/mL]; *—vs. NP_MC group, *p* < 0.001; #—vs PR_MC, *p* < 0.001; (**D**) serum total ghrelin levels [pg/mL]; *—vs. NP_MC group, *p* < 0.001; #—vs PR_MC, *p* < 0.001.

The experiment did not significantly affect the condition of the animals (Figure 5B), because no significant variation was found in the weight of animals (ANOVA: F(3,40) = 0.69; *p* = 0.5589) or in the total fluid intake (ANOVA: F(3,40) = 1.11; *p* = 0.3573).

In this experiment, the effect of PUE was also measured on active and total ghrelin levels in NP and PR rats (Figure 5C,D). On the basis of the obtained results, a statistically significant variability between the groups was observed for both active ghrelin (ANOVA: F(3,37) = 6.96; *p* = 0.0007) and a total one (ANOVA: F(3,34) = 4.82; *p* = 0.0066).

Furthermore, the statistical analysis revealed that PR_MC animals had significantly lower levels of active and total ghrelin (*p* < 0.001) relative to the corresponding NP_MC group (Figure 5C,D). The administration of PUE to PR animals (PR_PUE) increased the concentration of both forms of ghrelin in relation to PR_MC animals (*p* < 0.001) (Figure 5C,D). Such an effect of this isoflavonoid was not observed in NP_PUE rats in relation to NP_MC animals (*p* > 0.05).

### 2.5. AC Repeated Administration Effects on Alcohol Drinking Behavior and Ghrelin Levels

In another experiment, the effect of repeated (28×) intragastric (p.o.) administration of AC at a dose of 500 mg/kg on the amount of EtOH consumed in Wistar rats (*n* = 36) was observed. The rats were previously divided into preferring alcohol drinking (PR; *n* = 18) and non-preferring (NP; *n* = 18) groups. Later, they were randomly divided into four groups containing nine animals each (NP_MC, NP_AC, PR_MC, PR_AC).

It was observed that AC significantly influenced the variability of the amount of EtOH consumed (ANOVA: F(3,30) = 10.8; *p* = 0.0001). Further analysis showed that PR_MC animals drank much more alcohol compared to NP_MC animals (*p* < 0.001) (Figure 6A). AC administration significantly reduced the EtOH consumption in preferring animals (PR_AC) when compared to the PR_MC group (*p* < 0.001) (Figure 6A); in NP rats, no significant reduction in EtOH intake was observed (NP_AC vs. NP_MC, *p* > 0.05). At the same time, it was found that the administration of AC did not affect the condition of the animals in this experiment (Figure 6B), because there was no variation in the weight of the animals (ANOVA: F(3,30) = 0.316; *p* = 0.8196) or the total fluid intake (ANOVA: F(3,30) = 0.226; *p* = 0.8773).

In the same experiment, it was found that AC administration significantly influenced the variability of the obtained results for active (ANOVA: F(3,30) = 8.05; *p* = 0.0006) and total ghrelin levels (ANOVA: F(3,30) = 6.87; *p* = 0.0011). The level of active ghrelin, significantly lowered by alcohol consumption in PR rats (PR_MC vs. NP_MC; *p* < 0.001), increased to the control value (observed in NP rats) by AC administration (*p* < 0.001) (Figure 6C,D). Similarly, AC administration caused an increase in the level of total ghrelin in PR animals (AC_PR) when compared with the PR_MC group (*p* < 0.001). Such an effect was not seen in NP rats.

### 2.6. NAL Repeated Administration Effects on Alcohol Drinking Behavior and Ghrelin Levels

In the next experiment, the effect of repeated (28×) intraperitoneal (i.p.) administration of NAL at a dose of 0.1 mg/kg on the amount of EtOH consumed in Wistar rats (*n* = 28) was observed. The rats were previously divided into preferring alcohol drinking (PR; *n* = 14) and non-preferring (NP; *n* = 14) groups. Later, they were randomly divided into four groups containing seven animals each (NP_H_2_O, NP_NAL, PR_H_2_O, PR_NAL).

It was observed that NAL significantly influenced the variability of the amount of EtOH consumed (ANOVA: F(3,28) = 13.3; *p* = 0.0000). Further analysis showed that PR_H_2_O animals drank much more alcohol compared to NP_H_2_O animals (*p* < 0.001) (Figure 7A). The administration of NAL reduced the amount of EtOH consumed by preferring animals (PR_NAL) compared to the control preferring rats (PR_H_2_O, *p* < 0.001), whereas the drug did not affect the alcohol intake in NP animals (NP_NAL vs. NP_H_2_O, *p* > 0.05) (Figure 7A).

At the same time, it was found that the administration of NAL did not change the condition of the animals in this experiment (Figure 7B), as there was no variation in the weight of the animals (ANOVA: F(3,28) = 0.60; *p* = 0.6208), or the total fluid intake (ANOVA: F(3,28) = 1.07; *p* = 0.3789).

In this experiment, it was noted that the administration of NAL significantly influenced the changes in active (ANOVA: F(3,28) = 30.7; *p* = 0.0000) and total ghrelin concentrations (ANOVA: F(3,28) = 21.5; *p* = 0.0000). Further analysis showed that NAL significantly affected and caused an increase in the levels of both forms of ghrelin in PR animals (PR_NAL) when compared with control preferring rats (PR_H_2_O, *p* < 0.001), leading to values slightly higher than those observed in control NP animals (NP_H_2_O) (Figure 7C,D). Such an effect of NAL was not seen in NP animals.

### 2.7. KU Effects on EtOH Tolerance Development and Ghrelin Levels

This experiment was conducted in alcohol Warsaw high-preferring (WHP) rats. The effect of repeated once a day for 9 consecutive days (9×) administration of KU (500 mg/kg, p.o.) on the development of EtOH tolerance after repeated (9×) administration of EtOH (3.0 g/kg, i.p.) in comparison to H_2_O-treated animals was investigated. Selected rats (*n* = 32) were randomly divided into four groups containing eight animals each (H_2_O_MC, H_2_O_KU, Et_MC, Et_KU).

#### 2.7.1. EtOH Tolerance

The aim of this experiment was to evaluate the effect of KU, measured in the 30th, 60th, 90th minute after the administration of EtOH (3 g/kg, i.p.), on changes in rectal rat temperature. It should be stressed that at the beginning of the experiment (time t = 0), the rats did not differ in their baseline rectal temperature (38.2 ± 0.07 °C; ANOVA: F(3,22) = 1.92; *p* = 0.1488). In the 30th, 60th, and 90th minute, while examining the changes in temperature under the influence of KU and EtOH administration, significant variabilities were found. The obtained results of statistical calculations were as follows: in the 30th minute (main effect-temperature change: EtOH and KU, ANOVA with replication: F(3,22) = 40.0; *p* = 0.0000); effect of 9-day consecutive EtOH administration, ANOVA with replication: F(3,66) = 19.2; *p* = 0.0000); interaction between main effect and sequential administration effect: ANOVA with replication: F(9,66) = 5.03; *p* = 0.0004), in 60th minute (main effect-temperature change: EtOH and KU, ANOVA with replication: F(3,22) = 57.4; *p* = 0.0000; effect of 9-day consecutive EtOH administration, ANOVA with replication: F(3,66) = 12.41; *p* = 0.0000, interaction between main effect and sequential administration effect, ANOVA with replication: F(9,66) = 7.73; *p* = 0.0000), in the 90th minute (main effect-temperature change: EtOH and KU, ANOVA with replications: F(3,22) = 15.7; *p* = 0.0000; effect of 9-day consecutive EtOH administration, ANOVA with replications: F(3,66) = 12.1; *p* = 0.0000, interaction between main effect and sequential administration effect (ANOVA with replications: F(9,66) = 5.29; *p* = 0.0000).

Further statistical analysis, using the post hoc test, showed that on the first day, significant differences (*p* < 0.001) between the Et_MC and H_2_O_MC groups were found when measuring the temperature in the 30^th^, 60^th^, and 90^th^ minute after EtOH administration (Figure 8A–C). Moreover, on the third day of the experiment, there was also a significant difference between Et_MC and H_2_O_MC in the 60th minute (*p* < 0.001) (Figure 8B). On the remaining days, i.e., the fifth and eighth days, for all time points and on the third day (30 and 90 min after EtOH administration), the measured temperature differences between the groups receiving EtOH and H_2_O did not reach statistical significance (Figure 8A,C). When examining the effect of KU administration, it was found that in the animals from the Et_KU group, in the 30th minute on days three, five and eight, the body temperature was significantly lower than in the Et_MC group (*p* < 0.0001) (Figure 8A). A similar effect was observed for the measurement in the 60th minute (Figure 8B), where a statistically significant difference was recorded already on the first day of measurement and remained significant on all following days vs. the Et_MC group (*p* < 0.0001). In the 90th minute, the KU effect weakens and is significant only on the third and fifth days in relation to the Et_MC group (*p* < 0.0001) (Figure 8C).

Summarizing, 30 min after EtOH administration, the body temperature of animals treated with EtOH (Et_MC) decreased compared to the absolute control group (H_2_O_MC), whereas in the group receiving only KU (H_2_O_KU), there was no effect on temperature of KU alone (vs. H_2_O_MC), while the effect in the Et_KU group is clearly marked (Figure 8A). Similarly, 60 min after EtOH administration there is still a body temperature decrease in Et_MC animals compared to the control group (H_2_O_MC); for the Et_KU group the effect of the temperature drop remains strongly marked (Figure 8B). Moreover, 90 min after the EtOH administration, the body temperature also remained lower than in the control group (H_2_O_MC), and the values for the Et_KU group, as in the previous time intervals, remained the lowest (Figure 8C).

#### 2.7.2. Ghrelin Levels

In the experiment, significant group variability was also observed both for the active (ANOVA: F(3,22) = 5.36; *p* = 0.007) (Figure 8D) and total ghrelin levels (ANOVA: F(3,22) = 20.5; *p* = 0.0000) (Figure 8E). The post hoc analysis showed that after 9 days of EtOH administration in the Et_MC group, a significant decrease in the active (*p* < 0.05) and total (*p* < 0.001) ghrelin values in relation to the H_2_O_MC group was observed. In the group of animals receiving both EtOH and KU (Et_KU), a strong, statistically significant decrease in the level of active and total ghrelin was found in comparison to the control animals from the Et_MC group (*p* < 0.001) (Figure 8D,E).

#### 2.7.3. Body Weight

At the same time, it was noticed that the experiment did not significantly affect the body weight of the animals used (ANOVA: F(3,27) = 1.46; *p* = 0.2480) (Figure 8F).

## 3. Discussion

### 3.1. EtOH Intake

In this study, the results of five independent experiments related to the administration of KU, PUE, DAI, AC, and NAL on ghrelin levels in a selected model of free-choice alcohol consumption by rats were analyzed. This study was based on the assumption that the models of increased free-choice alcohol consumption would be used, obtained in Wistar animals using the induction model previously modified [47]. The introduction of this model and, especially, the induction and withdrawal syndrome that lasted 14 days, resulted in the appearance of a clear reflex of preference and led to the emergence of two extreme groups in terms of EtOH preferences: rats named non-preferring (NP) and preferring (PR) to drink alcohol by free choice, with significant differences in the amounts of consumed EtOH. This is consistent with the conclusions of other authors who emphasized that the use of such a procedure, especially the 14-day interval, allows to obtain the above mentioned NP and PR rats [47,54,55,56]. Two withdrawal periods, used in the experiments during which the rats only drank water, mimics the human pattern of drinking alcohol, due to the fact that an increased percentage of preference animals consumed significant amounts of alcohol [57,58]. In our model, approximately 20% of animals drank high amounts of EtOH and 20% drank small amounts. This is consistent with numerous studies which found rats to be rather resistant to alcohol addiction [58,59,60,61].

In drug-related studies on animals drinking alcohol, AC in a dose of 500 mg/kg, p.o. was used (28×), as in previous studies [62]. AC was used here as a standard drug with a recognized anti-alcohol effect [63,64,65,66,67,68,69], although the exact mechanism of its action is still under investigation [70,71]. As a result of the drug activity, free-choice alcohol consumption was reduced by 30% in PR rats, as compared to the corresponding control animals that did not receive the drug. Thus, the effectiveness of AC under model conditions was confirmed, which is consistent with other studies [62,72,73]. In a subsequent experiment, NAL was repeatedly administered (28×) per dose 0.1 mg/kg, i.p., to NP and PR rats, also as in our previous studies [74]. The effect of this standard drug on the drinking reflex was observed, and it was found that in the group of PR animals the amount of EtOH consumed decreased threefold compared to the placebo group, which is in line with other studies and human observations [75,76,77].

Kudzu—a raw material of plant origin, used in the present study as an extract (KU), is by its nature a mixture of various chemical compounds [78,79]. The use of kudzu as a substance of plant origin is noted in traditional medicine, as well as in more recent studies [10,17,80]. The isoflavonoids are the compounds that primarily determine the anti-alcoholic pharmacological activity of KU. Knowing the composition of the extract (Table 1), it was essential to verify whether the effect observed after the administration of KU alone, on drinking alcohol in free-choice procedure, would be similar to the effects of the isolated compounds, e.g., DAI and PUE, administered in separate experiments. The extract used in this study, with the content of PUE and DAI, was similar to one applied by other authors (in their studies on the influence of KU on the alcohol drinking behavior of rats) who administered KU with contents of 150 mg/g of PUE and 13 mg/g of DAI [79]. In our study, KU was administered in a dose of 500 mg/kg, p.o., for 28 days, according to data from other studies [78,81], where the authors noticed that KU in this dose significantly reduced the amount of alcohol drunk without affecting the total fluid intake or inducing weight changes. However, they also found that higher doses of KU (750 and 1000 mg/kg) did not result in a stronger effect on the alcohol reflex test but lowered the total amount of fluids consumed [78].

The free-choice alcohol drinking performance of KU-treated animals showed clear changes, expressed by a sixfold reduction in the amount of alcohol consumed by PR rats (PR_KU) when compared to vehicle-treated animals (PR_MC). It was consistent with the results of studies by other authors in which KU had a strong ability to inhibit the drinking reflex [10,78,80,81,82]. However, the KU mechanism of action is not obvious, since there were some observations that the isoflavones from KU suppressed alcohol drinking without entering the brain [78] and none of the compounds changed the activities of liver alcohol dehydrogenase (ADH) and aldehyde dehydrogenase (ALDH) [15]. On the contrary, the data shows that KU (*P. lobata* root extract produced in a special way called PLF) intravenous administration in doses of 80 and 160 mg/kg resulted in an occurrence (among others) of PUE in the brain, using an LC-MS/MS method in micro dialysate from striatal extracellular fluid of rats [83]. These authors also suggested that KU, especially in a lower dose (so-called optimal), promoted dopamine metabolism and inhibited serotonin metabolism without an influence on glutamine level. As for KU’s peripheral mechanism of action, there is also data showing that PUE (in a very high dose of 500 mg/kg, p.o.) can elevate ADH activity [84] and/or DAI can inhibit a mitochondrial type 2 aldehyde dehydrogenase action (ALDH-2) [14]. Hence, it can be inferred that the mechanism of KU activity requires further detailed exploration and clarification.

In the experiment on the influence of DAI on the drinking reflex, a dose of 40 mg/kg p.o. was applied. It was found that repeated administration of DAI (28×) reduced the amount of alcohol drunk by about 20% in animals belonging to the PR group, which suggests that DAI has an inhibitory effect on drinking EtOH in a free-choice procedure. It is known that the bioavailability of DAI administered per se is about 10 times lower than that administered in the same dose in the form of *P. lobata* root extract. Therefore, in our experiment, the dose of this compound was slightly higher than that, resulting directly from the mathematical calculation of the content in the investigated extract, also taking into account that DAI may have a reduced bioavailability [10,82]. Hence, the average dose of this flavonoid was chosen out of those used by other authors [85], knowing that the DAI administration at a much higher dose on the amount of alcohol drunk by rats (100 mg/kg, p.o.) [16] did not differ from our results in the applied dose. Moreover, in some studies DAI (30 mg/kg, p.o.) was administered in the form of pellets with the diet, although this procedure could cause an increase in bioavailability of this isoflavonoid [15]. Regardless of the above, it was also suggested that the diet had no effect on the activity of DAI as the substance reducing the EtOH consumption in animals [86].

PUE was the second isoflavonoid investigated for the effect of drinking EtOH in the free-choice procedure. The dose of this compound (150 mg/kg, p.o.) was selected on the basis of data from the literature [15,34,79,81], taking into account the typical percentage of isoflavonoids in the raw root material [87] and in different *P. lobata* extracts [84,88]. PUE was administered for 28 days according to the same schedule as the other substances used in our experiments. The obtained data showed that repeated administration of PUE decreased by half the amount of alcohol drunk in the PR control group (PR_MC), which is consistent with the results of other authors [15,79,81].

Summarizing, the analysis of our research on the use of KU, DAI, and PUE, as potential compounds with the effect of reducing alcohol drinking, showed that the strongest effect is produced by KU, which reduced EtOH drinking sixfold, while DAI and PUE yielded only 20% and 50% reductions, respectively. These observations suggest that it is not only the activity of the isoflavonoids present in KU that is responsible for the action of reducing drinking, as previously shown [15,81], but also other substances found in the extract, which may facilitate the absorption of DAI and PUE as active compounds, and/or influence the extract’s properties through other mechanisms. The verification of this hypothesis requires more in-depth investigation.

### 3.2. Tolerance

In our study on the alcohol tolerance development in WHP animals, daily intraperitoneal administration of EtOH (3 g/kg/day) led to noticeably rapid development, the effect already found on the third day of the experiment. This is in line with our previous similar experiments on Wistar animals [89,90,91]. Since KU induced lowering of EtOH consumption (see Section 2.7.1), it was decided to check its possible influence on the alcohol tolerance development. It was found that KU inhibited the development of EtOH tolerance, which was most apparently observed 60 min after the alcohol administration. The obtained results of this KU action are very interesting, especially because the development of such tolerance often precedes or accompanies the development of the alcohol addiction process [92,93]. Therefore, this profile of KU activity is crucial, and may constitute an option for its preventive use in groups of patients with an increased risk of alcoholism.

### 3.3. Ghrelin Level

In all experiments with the prolonged treatment with KU, DAI, PUE, AC, and NAL it was shown that only EtOH drinking produced a decline in both active and total ghrelin level, especially in PR rats. The nature of the relationship between ghrelin and alcohol is complex. However, as already mentioned in the introduction, drinking alcohol reduces the level of this peptide in the blood in humans [51,52] and in animal models, as was the subject of our previous research [47] and which was also shown in this work. This correlated with increased GHS-R1A gene expression in nucleus accumbens, ventral tegmental area, amygdala, prefrontal cortex, and hippocampus in AA rats [43,94]. However recently, it was found that alcohol does not act directly with the ghrelin system, although alcohol decreases peripheral ghrelin concentrations in vivo (also in humans), but not in proportion to alcohol’s caloric value or through direct interaction with ghrelin-secreting gastric mucosal cells, the ghrelin receptor, or the ghrelin-O-acyltransferase (GOAT) enzyme [95]. On the other hand, the administration of exogenous ghrelin has a more complex nature, as it leads to an increased alcohol supply since it has been proven in alcoholics who experienced increases in alcohol self-administration after administration of ghrelin in the form of an infusion [96]. However, further considerations in this field go beyond the scope of this work.

Repeated AC administration led to compensating for the above observation by increasing the level of ghrelin to the values found in control animals (NP). The mechanism by which the studied drug may alter ghrelin levels remains to date is still unknown. It is proven that blocking the NMDA receptor in the mesolimbic system (VTA) inhibits the increase in motor activity of animals observed after ghrelin administration [44]. However, it is unclear to what extent this mechanism is related to the effects of AC on the levels of both forms of ghrelin in our study. It was found that changes in ghrelin levels were noticeably inversely related to the amount of alcohol drunk by PR animals, which is in line with other observations [36,47]. While the exact mechanism of this relationship is unclear, it appears to be mainly due to the amount of available alcohol consumed by the PR rats. It is true that AC can lower the expression of the ghrelin gene in mice in the frontal cortex [97], but these studies were conducted without alcohol, hence the conclusions drawn from them cannot be easily applied to the studies in our work.

In the course of the study on the NAL influence on the free-choice alcohol drinking reflex in NP and PR rats, it was found that the drug had a normalizing effect on the reduced values of both ghrelin forms in PR rats, which was not observed in NP animals. This effect, as in the case of AC administration, is inversely related to the amount of alcohol drunk in the conditions of EtOH choice by PR animals. The nature of NAL’s effect on the increase in ghrelin levels, with the simultaneous abolition of alcohol drinking, is probably related to the action of alcohol itself. However, more complicated mechanisms related to the action of the ghrelin receptor on the activity of POMC and CART—peptides involved in the development of addiction—cannot be ruled out [98]. Nevertheless, it is not clear how ghrelin excites POMC neurons since there is some data suggesting that this phenomenon occurs in a different way than via activation of the ghrelin receptor coupled with excitation of agouti-related protein/neuropeptide Y [99].

In another experiment, PR and NP animals were given long-term KU. It was found that, unlike other substances, KU in the presence of EtOH does not bring the ghrelin levels to the control values but to a further decrease in their values. Similarly, the experiment on induction of alcohol tolerance, and after nine KU administrations, caused an intensification of the decrease in ghrelin levels in relation to the group receiving EtOH alone. The observation of the group receiving only KU (without EtOH) showed no decrease in the level of ghrelin. It can be assumed that this situation might have been caused by the caloric value of the extract, as KU was not a pure, active chemical but an alcoholic plant extract consisting of fats, proteins and carbohydrates, the administration of which causes a reduction in ghrelin levels [100]. However, this mechanism does not occur in our studies because, as already mentioned, in the experiment on KU influence on the EtOH tolerance development, the KU and water combined administration did not affect the concentration of both ghrelin forms. Hence, it seems that the reason for the decrease in ghrelin concentrations after administration of KU, in the presence of alcohol, is the unknown mechanism of compounds interaction present in the *P. lobata* root extract.

The subsequent studies also examined the effect of isoflavonoids from KU, namely DAI and PUE, on ghrelin levels. Similarly to AC and NAL administration, ghrelin leveling effects were observed in PR animals that received these isoflavonoids. DAI administration normalized the concentrations of both ghrelin forms in relation to the concentrations observed for the control NP animals. However, DAI did not change the levels of this peptide in NP rats. In the case of PUE, the normalization of level was observed only for the total ghrelin, whereas the active ghrelin form increased both in PR and (not significantly) in NP rats in comparison to control values.

It should be emphasized that the observed changes in ghrelin concentrations, under the influence of alcohol and substances used in our study, are related to an unknown but rather specific alcohol effect. It is highly possible because no changes in the body mass of rats or total fluid intake were noticed throughout the entire experiment. Therefore, the alcohol–ghrelin interrelation is not simply linked to the impact on the hunger or appetite center, where there is a correlation between ghrelin level and the feeling of hunger [101]. Although the mechanism by which AC, NAL, and both isoflavonoids could affect ghrelin levels is unknown, it is strongly correlated with a clear reduction in alcohol consumption in PR rats.

## 4. Materials and Methods

### 4.1. Substances and Drugs

The following reagents, substances and drugs were used in our study: ethyl alcohol-95% rectified spirit (Polmos, Bielsko-Biała, Poland); methylcellulose (MC) (Sigma-Chemical CO., Tokyo, Japan); water for injection (H_2_O)—“Aqua pro iniectione” (Fresenius Kabi Polska Sp. z o.o., Kutno, Poland); acamprosate (AC)—333 mg film-coated tablets (Campral) (Merck Sante, Semoy, France); naltrexone (NAL)—substance (98%) (Sigma-Aldrich, Poznań, Poland); *Pueraria lobata* (KU)—EtOH extract from the root of kudzu (KU)—extractum siccum (P.L. Thomas and Co., Inc., Morristown, NJ, USA); daidzin (DAI) (98%)—Glycine mass. (L.) Merr. (Shaanxi Sciphar Biotechnology Co., Ltd., Nanjing, China); puerarin (PUE) (98%)—Glycine mass. (L.) Merr. (Shaanxi Sciphar Biotechnology Co., Ltd., Nanjing, China).

### 4.2. Determination of Isoflavones in KU

Isoflavonoids from KU were analyzed according to the monograph for “Kudzu vine root” from European Pharmacopeia 8th and similarly to our previous research [90]. Briefly, reference substances (daidzin, daidzein) and acetic acid were obtained from Sigma-Aldrich (Millipore Sigma, Burlington, MA, USA). Standardized “Kudzu vine root” dry extract was obtained by EDQM. Ethanol and acetonitrile were purchased from J. T. Baker (J. T. Baker, Phillipsburg, NJ, USA).

Sample test: Approximately 0.6 g of dry extract was placed in a 250 mL round-bottomed flask and extracted with 50.0 mL of 30% ethanol under the reflux condenser for 30 min. After cooling down, the sample was transferred to a 50 mL volumetric flask and filled up with 30% ethanol. Next, 1.0 mL of this solution was transferred to a 25 mL volumetric flask and filled up with 30% ethanol and filtered through the 0.45 µm filters. Reference solutions: Approximately 30 mg of standardized “Kudzu vine root” dry extract was placed in a 250 mL round-bottomed flask and extracted with 50.0 mL of 30% ethanol under the reflux condenser for 30 min. After cooling down, the solution was filtered through the 0.45 µm filters. Approximately 1 mg of daidzein was placed in a 10 mL volumetric flask and filled up with 30% ethanol. Similarly, approximately 1 mg of daidzin was placed in a 10 mL volumetric flask and filled up with 30% ethanol.

HPLC-DAD analysis: An identification of isoflavonoids was performed on Agilent 1100 (Agilent, Santa Clara, CA, USA). Compounds were identified on LiChrospher 100 RP-18e, 150 mm × 4 mm × 5μm (Merck, Darmstadt, Germany). The mixture of two mobile phases eluted the compounds: CH3COOH: water 0.1:99.9 (*v*/*v*) (phase A) and acetonitrile (phase B) were used in the gradient separation procedure. The separation was performed in the following conditions: 0 min—10% phase B, 16.5 min—29% phase B. The column temperature was 25 °C, the flow rate was 3.0 mL/min, the detection was at λ = 260 nm, and the injection—10 µL. The peaks from the sample were identified by comparing the retention time and UV-VIS spectra with peaks from standardized “Kudzu vine root” dry extract (relative retention time described in the monograph). Relative retention with reference was as follows: to puerarin—about 2.1 min; 3-hydroxypuerarin—about 0.7; 3-methoxypuerarin—about 1.09; 6-O″-D-xylosypuerarin—about 1.15; daidzin—about 1.4. The concentrations of all substances were expressed as puerarin.

### 4.3. Animals and Experimental Protocols

The study was performed on male Wistar rats. The animals were obtained from the laboratory animal facilities (Laboratory Animals Cultivation, Lipiec Zbigniew, Brwinów, Poland) and they were used in the part of our study on the effect of substances/drugs on alcohol intake behavior. In the experiment on the development of alcohol tolerance (part of study on the effect of KU on EtOH tolerance development) male alcohol Warsaw High-Preferring Wistar rats (WHP; 56–58 generation, from the Institute of Psychiatry and Neurology, Warsaw, Poland) with a high voluntary EtOH intake were used as in our previous study [47].

All animals were kept in separate plastic cages (35 (l) × 20 (w) × 13 (h) cm with stainless steel covers) in a room with a temperature of 20 ± 2 °C, 65–75% humidity, and reversed circadian cycle (7 p.m. to 7 a.m. light). The rats had free access to water (except for the induction period), standard laboratory diet (pellets, Labofeed B) (except for 12 h before decapitation), and in respective periods of the study—10% (*w/w*) EtOH solution.

In order to select the alcohol-preferring animals (PR) as well as animals without such preference (non-preferring—NP) the procedure used before was applied [47,74,102,103], with slight modifications. This procedure is outlined in a Figure 1.

Briefly, for the first 2 weeks, Wistar rats were forced to drink only EtOH solution (10% (*w/w*). During the next 2 weeks, the animals were presented with a free-choice paradigm between tap water and EtOH (first preference period). Alcohol and water intake for every rat was recorded daily for the last 7 days of the preference period. The volumes of EtOH intake were converted to a value in g/kg/day and expressed as a mean ± SEM for the seven-day record. Similarly, the total fluid intake (the sum of water intakes and EtOH solution) was also defined as g/kg/day for all the groups. The next step involved the two-week alcohol withdrawal period, and rats were only provided with tap water. On the first day of the withdrawal period (after the first preference period), the body weight of rats was recorded for recalculation of the amount of drinking EtOH or water/kg. For the following 4 weeks, animals had free access to water and EtOH solution (second preference period). Again, in the 4th week, another set of fluid intake measurements was conducted on a daily basis. The body weight of animals on the first day of the second withdrawal period (after the second preference period) was recorded. According to second set of fluid intake measurements, EtOH treated animals were divided into two groups: (1) rats with a mean intake of EtOH exceeding 50% of the total fluid, (>3.5 g/kg/day), i.e., ‘preferring’ (PR) rats, and (2) rats in which alcohol solution constituted less than 50% of total fluid intake, i.e., ‘non-preferring’ (NP) rats.

### 4.4. Measurements of the Animals’ Body Weight

The animals were weighed before each experiment. In the experiments with long-term alcohol consumption and administration of substances or drugs, the measurements of body weight were taken at the beginning of each subsequent cycle to correctly calculate the amount (dose) of EtOH and water consumed/kg of body weight. In the further analysis, it was also considered whether the substances, drugs and alcohol affected the body weight of the animals.

### 4.5. KU, DAI, PUE, AC, and NAL Administration

This study involved five independent experiments according to the same model, carried out on PR and NP animals. After the procedure of obtaining the PR and NP animals, they were divided depending on the experiment into appropriate groups. Each group received a given substance or drug and the appropriate vehicle (0.5% methylcellulose (MC) for KU, DAI, PUE, AC, or water for injection (H_2_O) for NAL) once a day for 28 consecutive days (see Figure 2).

The doses of individual substances, along with the information on the type of solvent for substances and the number of animals used in each group, are presented in Table 2.

The volumes of EtOH intake were converted to a value in g/kg/day and expressed as a mean ± SEM for each group during the last week of drug/substance treatment. Similarly, the total fluid intake (the sum of intakes of water and EtOH solution) was also expressed as g/kg/day for the groups during the last week of drug administration.

### 4.6. KU Administration and the Development of EtOH Tolerance

The next experiment was planned over KU influence on EtOH tolerance development, using animals with genetically conditioned preference for voluntary alcohol intake (WHP). The tolerance was established by a daily intraperitoneal administration of EtOH at a dose of 3 g/kg for 9 consecutive days, according to the model proposed by Crabbe et al. [104] with some modifications [89,90,91]. This procedure is outlined in Figure 3.

Briefly, the rats were weighed and randomly divided into four groups: two groups with KU (500 mg/kg, p.o.) and two control groups which were intragastrically treated with 0.5% methylcellulose (MC) in complementary volumes. One hour after the KU administration, the rats were intraperitoneally injected with 30% EtOH at a dose of 3 g/kg or with water at complementary volume, respectively. Such a schedule was repeated for 9 consecutive days in order to develop alcohol tolerance in animals. On the first, third, fifth and eighth days, the animals’ body temperatures were measured to evaluate the hypothermic action of EtOH: once before the treatment (t = 0) (at the beginning of the experiment) and at three time points after the alcohol injection (30, 60, and 90 min after EtOH). The measurements were performed using a calibrated rectal electronic thermometer (TTK-3011, Temed, Zabrze, Poland). The probe of the apparatus was inserted into rectum for 20 s, which enabled establishing the temperature value on the scale. On the 9th day of the experiment, all animals were sacrificed by decapitation and peripheral blood was immediately collected for future analysis.

### 4.7. Serum Sampling and Protection for Downstream Analyzes

The animals subjected to measurement of ghrelin levels did not receive any food in the last 12 h before sacrifice in order to avoid the influence of food on ghrelin levels. After decapitation, blood samples (5 mL) were immediately taken on 1 mM ethylenediaminetetraacetic acid (EDTA, Sigma-Aldrich, Poznań, Poland) (500 µL) to avoid clotting, and on 100 µL of 1% p-hydroxymercuribenzoic acid (98%, Sigma-Aldrich, St. Louis, MO, USA) to prevent protein degradation by proteases. Next, the blood was centrifuged for 15 min (4500 r.p.m.) at a temperature of 4 °C. The acquired serum in the amount of 1.0 mL was secured by addition of 100 µL 1 M HCl to stabilize the sample and left to freeze at −80 °C for further measurements.

### 4.8. Measuring the Total and Active Ghrelin Levels

The measurement of total ghrelin concentration in the serum was determined using highly sensitive enzyme linked immunosorbent assay (ELISA) method and the commercially available kit (Phoenix Pharmaceuticals, Inc., Belmont, CA, USA) with 10 pg/mL detection limit.

The determination of active ghrelin (acylated) level in the serum was carried out by the ELISA method using the commercially available kit (SPI-Bio, Massy Cedex, France) with 1 pg/mL detection limit. All measurements were performed using the TECAN Sunrise ELISA plate reader (TECAN Austria Ges.m.b.H., Salzburg, Austria) and all stages of quantifications were carried out in accordance with the instructions attached in the form of a manufacturer’s protocol.

### 4.9. Statistical Analysis

The obtained values of repetitions in the groups were averaged and presented as arithmetic means ± SEM. Statistical calculations were performed using one-way analysis of variance (ANOVA) or ANOVA with replication and Fisher’s least significant difference post hoc test using the Statistica 13 program. The *p*-values < 0.05 were considered significant.

## 5. Conclusions

In conclusion, there was a proportional relationship between the anti-alcohol effect (reducing the alcohol-drinking reflex) of drugs (AC and NAL) or isoflavonoids (DAI and PUE) and the normalization in the levels of both ghrelin forms (active and total), which was still not observed for the effect of KU. Based on the above, it could be assumed that the peripheral concentration of both ghrelin forms may play an intriguing role as an indicator of alcohol-induced behavior, especially when controlling the effectiveness of currently used drugs. Nevertheless, this indicator should be treated with caution due to the fact that there are different alcohol intake vs. ghrelin level relationships while applying KU and its isoflavonoids. It is essential to stress that the action of KU on the ghrelin level is specific, and is not linked to its caloric value. Either way, the elucidation of KU-induced shift in ghrelin levels in the presence of EtOH requires further detailed study.

## Data Availability

The data presented in this study are available upon request from the corresponding authors.

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
