# Peer review of "Differential Influence of Pueraria lobata Root Extract and Its Main Isoflavones on Ghrelin Levels in Alcohol-Treated Rats"

_pharmaceuticals, 2021, doi:10.3390/ph15010025_

Round 1
Reviewer 1 Report
The manuscript "Differential influence of Pueraria lobata root extract and its main isoflavones on ghrelin levels in alcohol-treated rats" by Szulc M. et al. is devoted to the animal testing of different anti-alcoholic compounds and studying of ghrelin levels after anti-alcoholic treatment.
Ghrelin is a peptide hormone which regulate the food intake process and hunger sensing. As well as ghrelin could affect on the alcohol consumption: see, for example:
a. Farokhnia, M., Grodin, E.N., Lee, M.R. et al. Exogenous ghrelin administration increases alcohol self-administration and modulates brain functional activity in heavy-drinking alcohol-dependent individuals. Mol Psychiatry 23, 2029–2038 (2018). https://doi.org/10.1038/mp.2017.226
b. Landgren, S. et al. Expression of the gene encoding the ghrelin receptor in rats selected for differential alcohol preference. Behav Brain Res, 221, 182-188 (2011). https://doi.org/10.1016/j.bbr.2011.03.003.
c. Szulc M, Mikolajczak PL, Geppert B, Wachowiak R, Dyr W, Bobkiewicz-Kozlowska T. Ethanol affects acylated and total ghrelin levels in peripheral blood of alcohol-dependent rats. Addict Biol 2013; 18: 689–701. https://doi.org/10.1111/adb.12025
Guess, the work could be accepted for publication after minor revisions:
- Make the all entries of "H2O" with subscript "H2O";
- Add the more detailed discussion about the link between ghrelin and alcohol intake, incl. mentioned references a-c.
Author Response
Response to Reviewer 1
On behalf of my co-authors, I would like to thank you for giving us the great opportunity to revise our manuscript, which is entitled "Differential influence of Pueraria lobata root extract and its main isoflavones on ghrelin levels in alcohol-treated rats". We highly appreciate the reviewers’ insightful and helpful comments on our manuscript. We have revised the manuscript accordingly. For convenience, the implemented changes have been marked up in the revised manuscript.
A point-by-point response to the comments are as follows:
Ghrelin is a peptide hormone which regulate the food intake process and hunger sensing. As well as ghrelin could affect on the alcohol consumption: see, for example: [a,b,c]
Response: We truly appreciate the reviewer’s suggestions. Regulation of hunger and satiety with ghrelin is its primary function. The correlation between the rewarding effect of food and the rewarding effect of addictive substances, including alcohol, underlies the assumption that ghrelin may play a significant role in addiction modulation and its concentration measurement may turn out to be an important addiction marker.
The revised texts are shown as follows: The works cited [a, b, c] are known to us, the proposed works will be added to the list of publications and quoted in the discussion (Farokhnia et al.). A more recent work by Landgren et al from 2012 is cited [44, new number 95] but the proposed work (2011) will also be cited. The work of Szulc et al, is our own publication from earlier research and is already cited [52, new number is 94].
Make the all entries of "H2O" with subscript "H2O";
Response: We appreciate the reviewer’s suggestion. The abbreviation used by us is not a chemical formula but simply a name of a group. However, we agree that for the clarity and correctness of the work, the entry should be corrected for H2O.
The revised texts are shown as follows: Throughout the work, the H2O will be changed to H2O.
Add the more detailed discussion about the link between ghrelin and alcohol intake, incl. mentioned references a-c.
Response: We really appreciate the reviewer’s careful reviews. The suggested text with references will be completed in the ghrelin discussion subsection.
The revised texts are shown as follows: (...)The nature of the relationship between ghrelin and alcohol is complex. It is known, however, that, as already mentioned in the introduction, drinking alcohol reduces the level of this peptide in the blood in humans [50,51] and, among other things, which was the subject of our previous research, in the animal model [94], which was also shown in this work. This correlates with increased GHS-R1A gene expression in nucleus accumbens, ventral tegmental area, amygdala, prefrontal cortex, and hippocampus in AA rats [44,95]. On the other hand, the administration of exogenous ghrelin has a more complex nature, as it leads to an increased alcohol supply since it has been proven in alcoholics who experienced increases in alcohol self-administration after administration of ghrelin in the form of an infusion [96]. However, further considerations in this field go beyond the phenomena that are the aim of this work. (...)
[94] Szulc M, Mikolajczak PL, Geppert B, Wachowiak R, Dyr W, Bobkiewicz-Kozlowska T. Ethanol affects acylated and total ghrelin levels in peripheral blood of alcohol-dependent rats. Addict Biol. 2013 Jul;18(4):689-701. doi: 10.1111/adb.12025. Epub 2013 Jan 14. PMID: 23311595.
[44] Landgren S, Engel JA, Hyytiä P, Zetterberg H, Blennow K, Jerlhag E. Expression of the gene encoding the ghrelin receptor in rats selected for differential alcohol preference. Behav Brain Res. 2011, 221(1):182-8. doi: 10.1016/j.bbr.2011.03.003
[96] Farokhnia M, Grodin EN, Lee MR, Oot EN, Blackburn AN, Stangl BL, Schwandt ML, Farinelli LA, Momenan R, Ramchandani VA, Leggio L. Exogenous ghrelin administration increases alcohol self-administration and modulates brain functional activity in heavy-drinking alcohol-dependent individuals. Mol Psychiatry. 2018;23(10):2029-2038. doi: 10.1038/mp.2017.226
We tried our best to improve the manuscript and made some other changes in the manuscript, which will not influence the content of the paper. The revised manuscript has been proofread by a professional language editing service.
Reviewer 2 Report
Excessive alcohol consumption and related problems like alcohol abuse, adverse health effects, alcohol-related crimes, accidents, etc. are widespread community challenges. Therefore, the search and development of new deaddiction molecules are highly desirable. In the present manuscript, the authors discussed the ‘Differential influence of Pueraria lobata root extract and its 2 main isoflavones on ghrelin levels in alcohol-treated rats’. The overall idea is very intriguing, but the scientific content and rationale for the study are not very encouraging. Therefore, the study must be more conclusive to meet the level of the prestigious journal ‘Pharmaceuticals’ and authors need to consider the following issues:
- Several studies have already established that Kudzu roots, their extracts and/or isoflavones (DAI, PUE, ) for alcohol reduction properties (ref. 10-20).
- The authors have focused on the variation in the ghrelin blood level with the administration of KU and isoflavones. However, the ghrelin blood level trend is neither in line with the existing drugs nor is systematically conclusive to prove the authors’ claims.
- The authors need to provide a suitable justification for the variation in ghrelin level and mechanism to relate between ghrelin level and alcohol consumption?
- The authors should explain how the kudzu root extract dose was selected and how the extract was standardized for the PUE and DAI content?
- Some other minor issues the author should consider are
- The introduction is quite long and should be concise.
- The relative concentration of these isoflavones present in the different parts of the plant and their variation in their efficacy could be studied?
- The mechanism should be represented in figure form for better understanding.
- Include the chromatogram for the studies performed on HPLC-DAD?
Author Response
Response to Reviewer 2
On behalf of my co-authors, I would like to thank you for giving us the great opportunity to revise our manuscript, which is entitled "Differential influence of Pueraria lobata root extract and its main isoflavones on ghrelin levels in alcohol-treated rats". We highly appreciate the reviewers’ insightful and helpful comments on our manuscript. We have revised the manuscript accordingly. For convenience, the implemented changes have been marked up in the revised manuscript.
A point-by-point response to their comments are as follows:
- Several studies have already established that Kudzu roots, their extracts and/or isoflavones (DAI, PUE, ) for alcohol reduction properties (ref. 10-20).
Response: We agree with the reviewer's remark. However, It should be emphasized that the aim of the research was not to carry out one more proof that kudzu and its isoflavones have alcohol-reducing properties, whether we mentioned it in the introduction (lines 44-67) and in the discussion (lines 448-452, 466-470, 487-490). The studies used were needed to determine the variability of ghrelin levels in the blood of rats in the conditions of the model used under the influence of the substances used (kudzu, isoflavones) and two drugs (acamprosate, naltrexone) clinically used to reduce the desire to drink alcohol.
- The authors have focused on the variation in the ghrelin blood level with the administration of KU and isoflavones. However, the ghrelin blood level trend is neither in line with the existing drugs nor is systematically conclusive to prove the authors’ claims.
Response: We agree with the first part of this question, the actual results obtained after administering kudzu are different than for other substances or drugs. We assume that this discrepancy may be of great importance in the mechanisms of addiction regulation. All these aspect was presented in the discussion (chapter „Ghrelin level”). Undoubtedly, on the basis of today's knowledge available to the authors, it is not possible to explain the different results obtained under the influence of kudzu. This data could therefore be the subject of possible further research.
- The authors need to provide a suitable justification for the variation in ghrelin level and mechanism to relate between ghrelin level and alcohol consumption?
Response: This aspect is mentioned in the work and in the answer to a question from another reviewer. The exact mechanisms by which alcohol affects changes in ghrelin levels is unknown and has been discussed in other studies that we also cited.
The revised text is shown as follows: (...)The nature of the relationship between ghrelin and alcohol is complex. It is known, however, that, as already mentioned in the introduction, drinking alcohol reduces the level of this peptide in the blood in humans [50-51] and, among other things, which was the subject of our previous research, in the animal model [52], also shown in this work. This correlated with increased GHS-R1A gene expression in nucleus accumbens, ventral tegmental area, amygdala, prefrontal cortex, and hippocampus in AA rats [Landgren et al. 2011, 2012]. However recently, it was found that alcohol does not act directly with the ghrelin system, although alcohol decreases peripheral ghrelin concentrations in vivo (also in humans), but not in proportion to alcohol’s caloric value or through direct interaction with ghrelin-secreting gastric mucosal cells, the ghrelin receptor, or the ghrelin-O-acyltransferase (GOAT) enzyme [Deschaine SL, et al. 2021].
Deschaine SL, Farokhnia M, Gregory-Flores A, Zallar LJ, You ZB, Sun H, Harvey DM, Marchette RCN, Tunstall BJ, Mani BK, Moose JE, Lee MR, Gardner E, Akhlaghi F, Roberto M, Hougland JL, Zigman JM, Koob GF, Vendruscolo LF, Leggio L. A closer look at alcohol-induced changes in the ghrelin system: novel insights from preclinical and clinical data. Addict Biol. 2021:e13033. doi: 10.1111/adb.13033.
- The authors should explain how the kudzu root extract dose was selected and how the extract was standardized for the PUE and DAI content?
Response: The answer to the question is included in the discussion, where the choice of kudzu dose was presented (lines 442-447) and the standardization of the extract used (lines 438-442).
5. Some other minor issues the author should consider are:
a. The introduction is quite long and should be concise.
Response: The size of the introduction is large due to the complexity of the topic. The authors have carefully analyzed the introduction and, for the clarity, the text has been corrected and shortened.
b.The relative concentration of these isoflavones present in the different parts of the plant and their variation in their efficacy could be studied?
Response: This problem is undoubtedly interesting and has been the subject of some other research as, for example:
*Keung (see Ref. 10),
*McGregor (McGregor NR. Pueraria lobata (Kudzu root) hangover remedies and acetaldehyde-associated neoplasm risk. Alcohol. 2007;41(7):469-78),
*Takano (Takano A, Kamiya T, Tsubata M, Ikeguchi M, Takagaki K, Kinjo J. Oral toxicological studies of pueraria flower extract: acute toxicity study in mice and subchronic toxicity study in rats. J Food Sci. 2013;78(11):T1814-21.),
*Duan (Duan H, Cheng M, Yang J, Lai C, Zha L, Hu Y, Peng H, Huang L. Qualitative analysis and the profiling of isoflavonoids in various tissues of Pueraria Lobata roots by ultra performance liquid chromatography quadrupole/time-of-flight-mass spectrometry and high performance liquid chromatography separation and ultraviolet-visible detection. Phcog Mag 2018;14:418-24),
*Son (Son E, Yoon JM, An BJ, Lee YM, Cha J, Chi GY, Kim DS. Comparison among Activities and Isoflavonoids from Pueraria thunbergiana Aerial Parts and Root. Molecules. 2019, ;24(5):912).
In our work we have focused only on the most common source of kudzu isoflavones, commercially available, kudzu root.
c. The mechanism should be represented in figure form for better understanding.
Response: We don't quite understand this remark. The mechanism by which alcohol change the ghrelin level is not clearly defined and therefore it is difficult to present it in a graphic form. On the other hand, the mechanism of action of drugs or kudzu and its isoflavones that reduce alcohol consumption, as the reviewer himself wanted to note in question 1 (which we also emphasize in our answer), has already been the subject of many studies.
d. Include the chromatogram for the studies performed on HPLC-DAD?
We would like to emphasize that the method used to determine isoflavones is widely known and used (see European Pharmacopeia 8th, ref, 90) and the inclusion of details such as the chromatogram seems unnecessary for this type of work in our opinion.
We tried our best to improve the manuscript and made some other changes in the manuscript, which will not influence the content of the paper. The revised manuscript has been proofread by a professional language editing service.
Reviewer 3 Report
In this manuscript, the authors studied the different influence of the crude extract of Pueraria lobata root and its major isoflavones on ghrelin levels in alcohol-treated rats. The result of this study indicated Pueraria lobata (kudzu) root extract (KU) can reduce ghrelin levels in alcohol-preferring rats which suggests it may have different mechanism for inhibiting the alcohol intake of the studied rats compared to its main isoflavones, acamprosate, and naltrexonethe which cause an increase in bothtotal and active ghrelin levels in peripheral blood serum. Overall, the discovery of this study is significant. Besides, the manuscript is well-written and easy to follow. Combined together, I recommend the manuscript to be accepted when the following comments are addressed.
Major
A figure with chemical structures of key components of KU described in table 1 should be provided. The same to acamprosate (AC) and naltrexone (NAL).
Minor
The molecular formula of water should have a subscript "2" of H atom. Please correct them in the main text and the figures of the manuscript.
Schemes 1-2: the oxygen atom in the water molecule should be represented by the letter ‘O’ rather than the number ‘0’, please correct the typos in the schemes 1-2.
Schemes 2-3: ‘colection’ should be corrected to collection.
Author Response
Response to Reviewer 3
On behalf of my co-authors, I would like to thank you for giving us the great opportunity to revise our manuscript, which is entitled "Differential influence of Pueraria lobata root extract and its main isoflavones on ghrelin levels in alcohol-treated rats". We highly appreciate the reviewers’ insightful and helpful comments on our manuscript. We have revised the manuscript accordingly. For convenience, the implemented changes have been marked up in the revised manuscript.
A point-by-point response to their comments are as follows:
Major
A figure with chemical structures of key components of KU described in table 1 should be provided. The same to acamprosate (AC) and naltrexone (NAL).
Response: this work is not of a chemical nature, but for the clarity of the work, chemical formulas of key compounds in this work (DAI, PUE, AC, NAL) have been added.
The revised texts are shown as follows: figure with chemical structures of main compounds have been added.
Minor
The molecular formula of water should have a subscript "2" of H atom. Please correct them in the main text and the figures of the manuscript.
Response: We appreciate the reviewer’s suggestion. The abbreviation used by us is not a chemical formula but simply a name of a group and its notation has been carefully prepared in this way. However, we agree that for the clarity and correctness of the work, the entry should be corrected for H2O.
The revised texts are shown as follows: Throughout the work, the H2O will be adjusted to H2O.
Schemes 1-2: the oxygen atom in the water molecule should be represented by the letter ‘O’ rather than the number ‘0’, please correct the typos in the schemes 1-2.
Response: Thank you for pointing this. Obviously it was a mistake. Letter “O” was corrected in all H2O.
The revised texts are shown as follows: Throughout the work, the H2O was corrected.
Schemes 2-3: ‘colection’ should be corrected to collection.
Response: Thank you for finding this error. It was corrected.
We tried our best to improve the manuscript and made some other changes in the manuscript, which will not influence the content of the paper. The revised manuscript has been proofread by a professional language editing service.
Round 2
Reviewer 2 Report
Authors' should provide a valid explanation for the contrasting results for KU vs ghrelin level.
Author Response
Authors' should provide a valid explanation for the contrasting results for KU vs ghrelin level.
In response, we would like to underline, as we have already mentioned in previous response to a similar comment from the reviewer, that on the basis of our knowledge, we are not able to explain the different effects of KU on ghrelin levels in the presence of alcohol compared to the drugs or isoflavonoids described in our work.
It seems that the administration of KU only (without alcohol) did not change the concentration of ghrelin in the blood of rats, which was presented in the part of the work on the possibility of inhibiting the development of ethanol tolerance by KU (Fig. 8). As shown only in the presence of ethanol, the levels of ghrelin decreased after administration of KU, which was not observed after administration of KU without alcohol. It follows that this effect is specific to the KU-ethanol interaction, which results in a reduction in ghrelin levels.
This is consistent with the results obtained in the section on the inhibitory effect of the free-choice drinking reflex in preferring rats, where after administering KU, although the amount of alcohol drunk decreases, the effect measured by ghrelin level is different than after administration of isoflavones or both drugs which we also wrote about in the discussion (lines 540-554).
It should be emphasized that the dose of KU used did not change the weight of animals in our experiment, which is in line with the results of other authors discussing this aspect of kudzu activity, due to its possible caloric content (see - discussion lines 425-430, 546-549).
It cannot be ruled out that this effect is related to the anti-stress effect of KU (some species of Pueraria show such properties - see Pramanik et al. 2010, [Pramanik SS, Sur TK, Debnath PK, Bhattacharyya D. Effect of Pueraria tuberosa tuber extract on chronic foot shock stress in Wistar rats. Nepal Med Coll J. 2010 Dec;12(4):234-8], because it is known that stress in most cases increases the level of cortisone, which leads to an increase in ghrelin levels, and a reduction in the level of cortisone leads to a decrease in the concentration of this peptide in the blood [Azzam et al. 2017]. The reduction in the level of cortisone was observed which occurred during the withdrawal from alcohol and was exacerbated by the administration of Puerariae flos extract [Jiang et al. 2021].
Jiang B, Yang W, Xiu Z, Zhang L, Ren X, Wang L, Chen L, Asakawa T. An in vivo explorative study to observe the protective effects of Puerariae flos extract on chronic ethanol exposure and withdrawal male mice. Biomed Pharmacother. 2021 May;137:111306. doi: 10.1016/j.biopha.2021.111306.
Azzam I, Gilad S, Limor R, Stern N, Greenman Y. Ghrelin stimulation by hypothalamic-pituitary-adrenal axis activation depends on increasing cortisol levels. Endocr Connect. 2017 Nov;6(8):847-855. doi: 10.1530/EC-17-0212.
But why should this effect occur only for KU in the presence of alcohol and give different results under the same conditions for the main components of KU? There is a hypothesis that is difficult to resolve on the basis of our present knowledge
It cannot be ruled out that this effect would be due to other components of KU, present in a smaller amount than its main components - isoflavons, but this aspect can only be resolved after undertaking appropriate research aimed at this hypothesis.
Such explanations are highly speculative, so we believe that they should not be included in the work. Authors' should provide a valid explanation for the contrasting results for KU vs ghrelin level. If not, this is a goal for further investigations as we mentioned already in discussion.